# Both In Situ and Circulating SLC3A2 Could Be Used as Prognostic Markers for Human Lung Squamous Cell Carcinoma and Lung Adenocarcinoma

**DOI:** 10.3390/cancers14215191

**Published:** 2022-10-22

**Authors:** Dahua Liu, Min An, Guimin Wen, Yanan Xing, Pu Xia

**Affiliations:** 1Biological Anthropology Institute, College of Basic Medical Science, Jinzhou Medical University, Jinzhou 121001, China; 2Department of Cardiology, Jinzhou Central Hospital, Jinzhou 121001, China; 3Department of Basic Nursing, College of Nursing, Jinzhou Medical University, Jinzhou 121001, China; 4Department of Surgical Oncology, First Affiliated Hospital of China Medical University, Shenyang 110001, China

**Keywords:** SLC3A2, lung squamous cell carcinoma, lung adenocarcinoma, prognosis, MEK/ERK pathway

## Abstract

**Simple Summary:**

With the continuous progress of diagnosis and treatment technology, the early diagnosis rate and survival rate of lung cancer have improved, but the incidence rate and mortality rate of lung cancer are still very high. Therefore, it has become an urgent problem to analyze the molecular mechanism of lung cancer and to determine the markers related to early diagnosis. SLC3A2 protein is a cell-surface marker that plays an important role in tumorigenesis and development, and it is expected to become a new target for the treatment of tumors. The in-depth study of SLC3A2 can provide a new molecular target for the early diagnosis, treatment, and prognosis of lung cancer.

**Abstract:**

SLC3A2, the heavy chain of the CD98 protein, is highly expressed in many cancers, including lung cancer. It can regulate the proliferation and the metastasis of cancer cells via the integrin signaling pathway. Liquid biopsy is a novel method for tumor diagnosis. The diagnostic or prognostic roles of serum SLC3A2 in lung cancer are still not clear. In this study, we analyzed SLC3A2 mRNA levels in human lung squamous cell carcinoma (LUSC) and lung adenocarcinoma (LUAD) using the TCGA database and serum SLC3A2 protein levels using ELISA. We confirmed high SLC3A2 levels in both the serum and tissue of LUAD and LUSC patients. Both serum and tissue SLC3A2 could be used as prognostic markers for overall LUAD and subgroups of LUSC patients. SLC3A2 induced tumorigenesis via the MEK/ERK signaling pathway in LUAD and LUSC cells.

## 1. Introduction

Primary bronchogenic carcinoma is a malignant tumor that originates from the trachea, bronchus, and lung [1,2,3]. Lung cancer is a bronchogenic cancer, examples of which include lung squamous cell carcinoma (LUSC), lung adenocarcinoma (LUAD), small-cell carcinoma, and large-cell carcinoma [4,5,6]. Lung cancer has a high incidence and mortality worldwide [7]. The pathogenesis of lung cancer has not been fully understood, but there is evidence that the occurrence of lung cancer is related to smoking, air pollution, occupational carcinogens, diet, genetics, and other factors [8,9,10]. Most patients with lung cancer are in an advanced stage when they go to hospital [11]. Early detection, diagnosis, and treatment of lung cancer are very important for improving the survival of patients [12].

CD98 is a heterodimer transmembrane glycoprotein that contains a glycosylated heavy chain (CD98hc, SLC3A2, 4F2, 4F2hc) and a nonglycosylated light chain (LATl, LAT2, XCT, etc.) [13]. It has two main functions: transporting amino acids and regulating the integrin signaling pathway [14]. The SLC3A2 transmembrane domain and intracellular structure interact with the integrin β1 and β3 subunits and participate in cell proliferation, migration, invasion, and adhesion regulated by the integrin signaling pathway [15]. SLC3A2 is highly expressed in LUSC [16] and LUAD [17]. SLC3A2 silencing in cancer cells reverts tumorigenesis, migration, and proliferation [18,19]. Serum proteins have been used as diagnostic and prognostic markers for cancers for more than thirty years [20]. As an auxiliary means, serum protein improves the accuracy of diagnosis, such as CEA for colorectal cancer [21]. However, to our knowledge, no previous studies have shown the diagnostic or prognostic roles of serum SLC3A2 in lung cancer.

In this study, we used the TCGA database to analyze the prognostic roles of SLC3A2 in LUAD and LUSC patients. We collected blood samples from LUAD and LUSC patients and checked the serum SLC3A2 protein levels of these patients. Finally, we validated the roles and mechanisms of SLC3A2 in LUAD and LUSC cells.

## 2. Materials and Methods

### 2.1. Bioinformatics Analysis

UALCAN (http://ualcan.path.uab.edu, accessed on 5 November 2020) is a platform for the in silico analysis of cancer transcriptome data from The Cancer Genome Atlas (TCGA) database [10]. The Cancer Genome Atlas (TCGA), a project jointly launched by the National Cancer Institute (NCI, National Cancer Institute of the United States) and the National Human Genome Research Institute (NHGRI, National Human Genome Research Institute of the United States) in 2006, contains clinical data, genomic variation, mRNA expression, miRNA expression, methylation, and other data of various tumor molecular subtypes, such as individual age, gender, or tumor stage [10]. We explored the relative expression of SLC3A2 in primary LUAD and LUSC tissues via UALCAN based on various tumor molecular subtypes and the clinicopathological features of the patients. Differences in transcriptional expression were compared using Students’ *t* test, and *p* < 0.01 was considered statistically significant.

### 2.2. Blood Samples

Blood samples were obtained from 66 LUAD and 52 LUSC patients who had received no chemotherapy or radiotherapy prior to resection at the Department of Thoracic Surgery, the First Hospital of China Medical University, between July 2013 and July 2015. All the patients signed informed consent. Control blood samples were obtained from 38 healthy individuals who were of similar age and who had similar daily lifestyles as the patients. This study was conducted according to the Helsinki Declaration of 1975 and was approved by the Ethics Committee of Liaoning Medical University (LMU20150012).

### 2.3. ELISA

Blood samples were collected in EDTA vacutainers and then centrifuged at 1500× *g* for 10 min. Serum SLC3A2 levels were measured using an enzyme-linked immunosorbent assay (ELISA) according to the manufacturer’s instructions (USCN Business Co., Ltd., Wuhan, China). Briefly, blood samples were added into 96-well plates. The plates were sealed with a cover and were incubated at 37 °C for 90 min. Then, the content of the plates was discarded, and the plates were washed two times. Next, 100 µL Biotin-labeled antibody working solution was added to each well, and the plates were incubated at 37 °C for 60 min. HRP-Streptavidin Conjugate (SABC) (100 µL) was added into each well, and the wells were then covered with a plate and incubated at 37 °C. After 30 min, 90 µL TMB Substrate was added into each well, and plates were incubated at 37 °C in the dark for 10–20 min. O.D. absorbances were read at 450 nm in a microplate reader (Thermo Fisher Scientific, Shanghai, China) immediately after adding the stop solution.

### 2.4. Cell Culture and SLC3A2 Silencing

The human lung adenocarcinoma cell line H1975 and human lung squamous cell carcinoma cell line H226 were purchased from the Shanghai Institute of Cell Biology, Chinese Academy of Sciences (Shanghai, China). Cells were maintained in a humidified cell incubator with 5% CO_2_ at 37 °C in DMEM supplemented with 10% FBS (KeyGEN, Nanjing, China). Cells were plated onto 6-well plates at a density of 3 × 105 cells per well. Cells at 60–70% confluency were transfected with SLC3A2 siRNA1, sense: GGACCUCACUCCCAACUAUTT, antisense: AUAGUUGGGGAGUGAGGUCCTT; SLC3A2 siRNA2, sense: CAGATCCTGAGCCTACTCGAA, antisense: TCCGTGTCATTCTGGACCTTA; and scrambled siRNA: AATTCTCCGAACGTGTCACGT (Qiagen, Shanghai, China).

### 2.5. Apoptosis Assay

Cells were collected and washed with PBS and then resuspended in AnnexinV-FITC and PI (0.5 μg/mL) (Apoptosis Detection Kit, KeyGEN) in the dark for 30 min. Then, the cells were immediately analyzed on a FACSCalibur flow cytometer (Becton Dickinson Medical Devices, Shanghai, China).

### 2.6. 5-Aza-2′-Deoxcytidine (5-AzaC) Treatment

Cells were seeded at a density of 1 × 10^5^ cells per well in six-well culture plates and treated with 5-aza-2′-deoxycytidine (Sigma-Aldrich, Saint Louis, MO, USA) at concentrations of 2.5 and 5 µM daily for 5 days.

### 2.7. Immunoblotting

Cells were lysed in RIPA Lysis Buffer (Beyotime Biotechnology, Shanghai, China) containing a protease inhibitor cocktail (Sigma-Aldrich). Proteins (40 μg per lane) were separated by 10% SDS-polyacrylamide gel electrophoresis and transferred to a nitrocellulose (NC) filter membrane (Beyotime Biotechnology). Primary antibodies were SLC3A2 (sc-59145, Santa Cruz Biotechnology; Santa Cruz, CA, USA), P-ERK (sc-7383; Santa Cruz), ERK (sc-271270; Santa Cruz), P-MEK (sc-271914; Santa Cruz), MEK (sc-6250; Santa Cruz), and GAPDH (sc-47724; Santa Cruz). The secondary antibody was alkaline phosphatase-conjugated mouse IgG (KeyGEN). Detection of the immune complexes was performed with the ECL Western blotting detection system (Amersham Biosciences, Piscataway, NJ, USA).

### 2.8. Statistical Analysis

GraphPad Prism 5 software (GraphPad Software Inc., San Diego, CA, USA) was used to analyze all experimental data and clinical data. One-way ANOVA was used for intergroup comparison, and Dunnett’s T3 test was used for intragroup comparison. The area under the ROC curve, sensitivity, and specificity were used to predict the diagnostic value of serum SLC3A2 for both LUAD and LUSC. Log-rank statistical analysis was used to compare the differences between Kaplan–Meier survival curves. *p* < 0.05 was statistically significant.

## 3. Results

### 3.1. SLC3A2 mRNA Levels and Roles in LUAD and LUSC Tissues

We compared the mRNA levels of SLC3A2 in LUAD and LUSC tissues by analyzing the TCGA database. The mRNA level of SLC3A2 was higher in both LUAD and LUSC tissues than in matched normal tissues (*p* < 0.05, Figure 1A and Figure 2A). SLC3A2 mRNA was highly expressed both in Stage 1–4 LUAD and LUSC tissues compared to in matched normal tissues (*p* < 0.05, Figure 1B and Figure 2B). There was no difference in the SLC3A2 mRNA level in LUAD and LUSC patients of different races and ages, except LUSC African American patients (Figure 1C,E and Figure 2C,E). No difference of SLC3A2 was found in male and famale LUAD patients (Figure 1D). But male LUSC patients expressed higher SLC3A2 than female patients (*p* < 0.05, Figure 2D). SLC3A2 mRNA also was highly expressed in both N 0–4 LUAD and LUSC tissues compared to in matched normal tissues (*p* < 0.05, Figure 1F and Figure 2F). SLC3A2 expression in LUAD and LUSC is based on histology subtypes (Figure 1G and Figure 2G). Smoking habits did not influence SLC3A2 expression in patients (Figure 1H and Figure 2H).

### 3.2. Methylation Status of SLC3A2 in LUAD and LUSC Tissues

Interestingly, we found that the SLC3A2 mRNA levels in lung cancer were associated with methylation. No significant differences were observed for SLC3A2 methylation in LUAD tissues and matched normal tissues (Figure 3A,B,F). However, hypermethylation was observed in normal tissues compared to in LUSC tissues (Figure 4A,B,F). Methylation status of SLC3A2 showed no big difference in race, gender, age and smocking habits of both LUAD and LUSC patients (Figure 3C,D,E,G and Figure 4C,D,E,G).

### 3.3. Tissue SLC3A2 Was a Prognostic Marker for Overall LUAD and Subgroups of LUSC Patients

High SLC3A2 expression was associated with a poor prognosis in LUAD patients (*p* = 0.044) (Figure 5A). The influence of SLC3A2 on the prognosis of LUAD patients was not related to race (*p* = 0.11) (Figure 5B), sex (*p* = 0.15) (Figure 5C), or smoking habits (*p* = 0.11) (Figure 5D). There was no significant difference in survival between the LUSC patients with high and low SLC3A2 expression (*p* = 0.56) (Figure 6A). However, female patients (*p* = 0.0033) (Figure 6B), African American patients (*p* = 0.0033) (Figure 6C), and nonsmoker patients (*p* = 0.0031) (Figure 6D) with high SCL3A2 expression had a lower survival probability than the patients with low expression.

### 3.4. Serum SLC3A2 Protein Levels and Roles in LUAD and LUSC Patients

We found that the serum SLC3A2 protein levels were higher in both LUAD and LUSC patients compared to in healthy volunteers using ELISA (Figure 7A). The ROC curves of SLC3A2 protein revealed strong discrimination between LUAD patients with an AUC of 0.936 (*p* < 0.0001, Figure 7B) and LUSC patients with an AUC of 0.923 (*p* < 0.0001, Figure 7C). The mean value of serum SLC3A2 proteins was used as the standard to divide the patients into high- and low-expression groups. Serum SLC3A2 protein was associated with tumor differentiation, lymphatic invasion, and venous invasion in both LUAD and LUSC patients (*p* < 0.05, Table 1). The associations of serum SLC3A2 protein with the clinicopathological characteristics of the LUAD and LUSC patients are summarized in Table 1. Kaplan–Meier analysis showed that the SLC3A2 protein level indicates poor prognosis of LUAD and LUSC patients (*p* < 0.05, Figure 7D,E).

### 3.5. Tumorigenesis Roles and Mechanisms of SLC3A2 in LUAD and LUSC Cells

After SLC3A2 knockdown, the apoptotic ratio of LUAD and LUSC cells was increased using Annexin V-FITC/PI double staining (Figure 8A). P-ERK and P-MEK were downregulated in SLC3A2 knockdown cells (Figure 8B). No significant changes in ERK or MEK were observed (Figure 8B). As mentioned above, bioinformatic analysis showed that SLC3A2 methylation happened in LUSC cancer tissues but not in LUAD tissues. We validated these results in LUAD and LUSC cells using 5-AzaC, a nucleoside-based DNA methyltransferase inhibitor. Consistent with tissues, we confirmed that SLC3A2 was increased slightly in both LUAD and LUSC cells after receiving 5-AzaC treatment (Figure 8C). In addition, we showed the correlation genes of SLC3A2 in both LUAD and LUSC tissues (Figure 9). These data can give us suggestions for studying the deep mechanisms of SLC3A2.

## 4. Discussion

The upregulation of SLC3A2 was correlated with tumorigenesis, metastasis, and metabolism [15]. SLC3A2 can also be used as a prognostic marker for human cancers [22,23,24,25]. Positive SLC3A2 expression predicts poor prognosis and increased recurrence of NSCLC patients [17]. In this study, we used the TCGA database to provide an overview of SLC3A2 expression in different subgroups of lung cancer patients. SLC3A2 was expressed at similar levels in LUAD and LUSC tissues of different stages but at low levels in matched normal tissues. These results mean SLC3A2 is a tumor initiation factor rather than a tumor development factor. Consistent with previous studies, high levels of SLC3A2 indicated a poor prognosis in LUAD and LUSC patients.

The main finding of this study is the correlation of serum SLC3A2 with the prognosis of LUAD and LUSC patients. To our knowledge, this is the first study that has shown the expression levels and roles of serum SLC3A2 in lung cancer patients. The emergence of noninvasive diagnostic technology, namely liquid biopsy, represents great progress in tumor diagnosis and treatment [26]. Fluid biopsy technology mainly includes the detection of free circulating tumor cells (CTCs), circulating tumor DNA (ctDNA), exosomes, and circulating proteins [27]. Compared to traditional tissue biopsy, fluid biopsy has the unique advantages of real-time dynamic detection, overcoming tumor heterogeneity, and providing comprehensive detection information [27]. In this study, we confirmed that serum SLC3A2 is associated with tumor differentiation, lymphatic invasion, and venous invasion in both LUAD and LUSC patients. Kaira et al. [16] found that high SLC3A2 expression in lung cancer tissues is related to lymphatic invasion. In addition, serum SCL3A2 can be used as prognostic marker for both LUAD and LUSC patients. These results indicate that both serum SLC3A2 and tissues SLC3A2 are related to the clinical features of LUAD and LUSC patients.

Finally, we validated the roles of SLC3A2 in LUAD and LUSC cells using small interfering RNA technology. After SLC3A2 knockdown, the apoptotic ratio was upregulated, and the MEK/ERK pathway was inhibited in both LUAD and LUSC cells. The MEK/ERK signaling pathway is a mitogen-activated protein kinase (MAPK) pathway [28,29,30]. Activation of the MEK/ERK signaling pathway can cause a protein kinase cascade reaction and transmit extracellular signals into cells [16]. The imbalance of the MEK/ERK signaling pathway plays an important role in tumorigenesis and development [16]. In our study, P-MEK and P-ERK were downregulated in both LUAD and LUSC cells after SLC3A2 knockdown; however, total MEK and ERK did not change. Furthermore, we found SLC3A2 methylation in LUSC cancer tissues using bioinformatic analysis, and this was validated in both LUSC and LUAD cells. SLC3A2 was increased slightly in both LUAD and LUSC cells after 5-AzaC treatment. This means that SLC3A2 methylation blocked its expression in normal lung tissues. It is a protective factor against LUAD and LUSC. Methylation is a reason that can partly explain the SCL3A2 expression in lung cancer tissues and cells. The expression of SLC3A2 is also regulated by other mechanisms, so the removal of promoter methylation can only slightly increase its expression.

Neuregulin 1 (NRG1), which belongs to the epidermal growth factor family, is a signal protein that mediates cell–cell interactions [31]. SLC3A2 is a fusion partner gene of NRG1 and forms a new fusion protein on the cell membrane [32]. The SLC3A2-NRG1 fusion gene has been widely found in lung cancer cells [32]. At present, afatinib, a targeted drug of NRG1, can effectively shrink tumors, prolonging the survival time of patients [33]. Zenocutuzumab is effective in patients with SLC3A2-NRG1 fusion-positive NSCLC [34]. We can also consider targeting drugs against SLC3A2 to interfere with the SLC3A2-NRG1 fusion gene to treat tumors. For instance, Anticalin (P3D11) is a newly designed protein that works against the extracellular domain of SLC3A2 [35].

## 5. Conclusions

In this study, we confirmed that both serum and tissue SLC3A2 could be used as prognostic markers for overall LUAD and subgroups of LUSC patients. In addition, SLC3A2 induced tumorigenesis via the MEK/ERK signaling pathway in lung cancer cells. All this evidence indicates that SLC3A2 has important application value in the early diagnosis, histological classification, clinical stage, prognosis, and efficacy monitoring of lung cancer. SLC3A2A is also expected to become a new target for the targeted treatment of lung cancer. In the future, we will design targeted small molecule drugs based on the protein structure of SLC3A2, and study anticancer drug efficacy from basic research to clinical application.

## Figures and Tables

**Figure 1 cancers-14-05191-f001:**
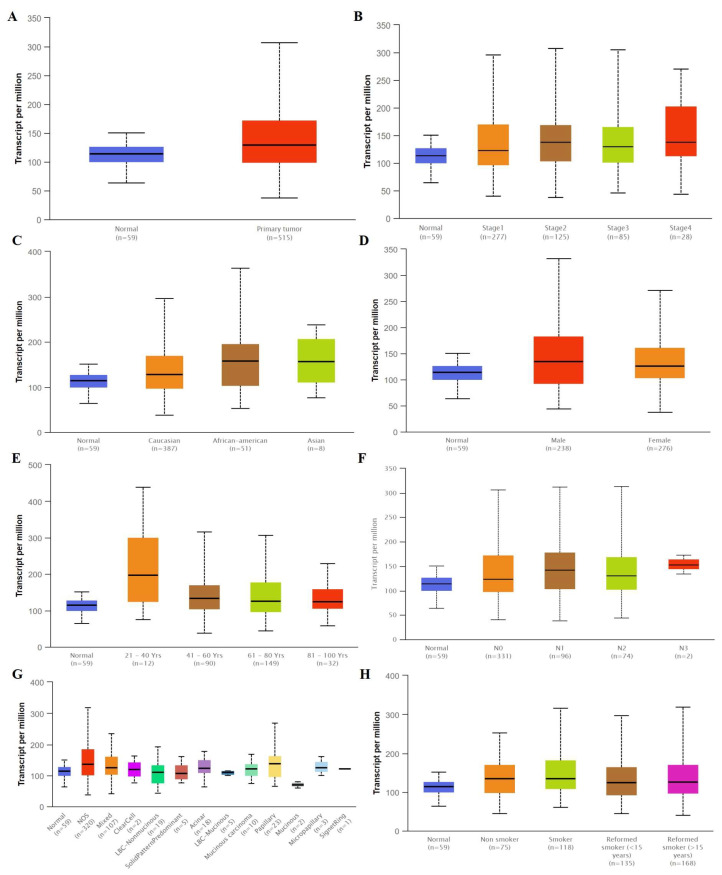
The mRNA level of SLC3A2 expressed in tumor tissues and in matched normal tissues from different subgroups of LUAD patients. (**A**) Sample type; (**B**) Stages; (**C**) Race; (**D**) Gender; (**E**) Age; (**F**) Nodel metastasis; (**G**) Tumor histology; (**H**) Smoking habits.

**Figure 2 cancers-14-05191-f002:**
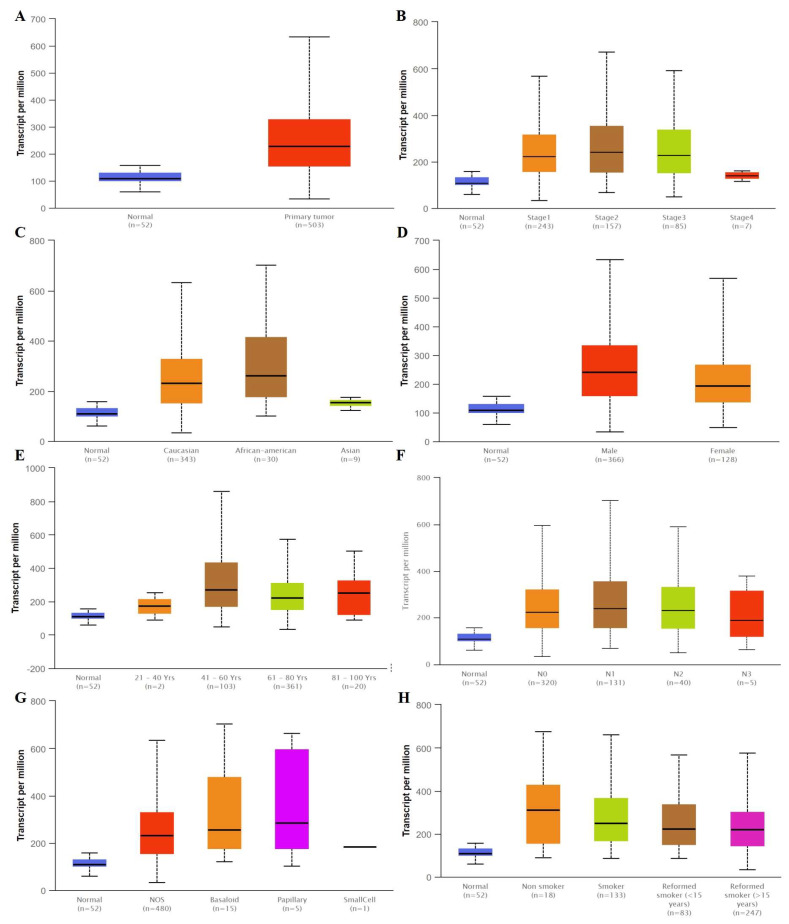
The mRNA level of SLC3A2 expressed in tumor tissues and in matched normal tissues from different subgroups of LUSC patients. (**A**) Sample type; (**B**) Stages; (**C**) Race; (**D**) Gender; (**E**) Age; (**F**) Nodel metastasis; (**G**) Tumor histology; (**H**) Smoking habits.

**Figure 3 cancers-14-05191-f003:**
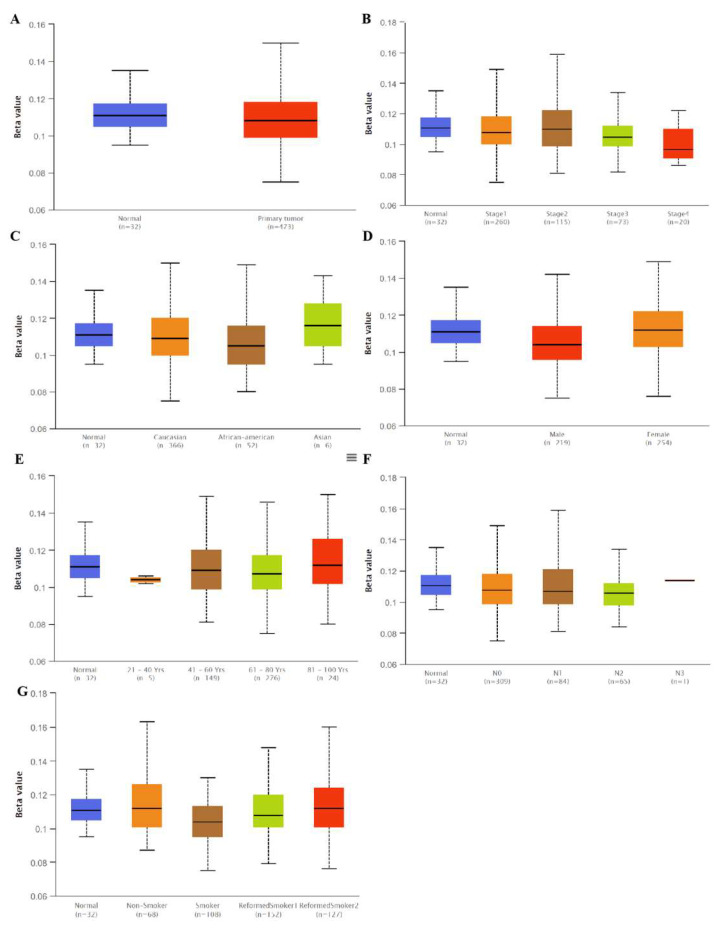
Methylation status of SLC3A2 in tumor tissues and in matched normal tissues from different subgroups of LUAD patients. (**A**) Sample type; (**B**) Stages; (**C**) Race; (**D**) Gender; (**E**) Age; (**F**) Nodel metastasis; (**G**) Smoking habits.

**Figure 4 cancers-14-05191-f004:**
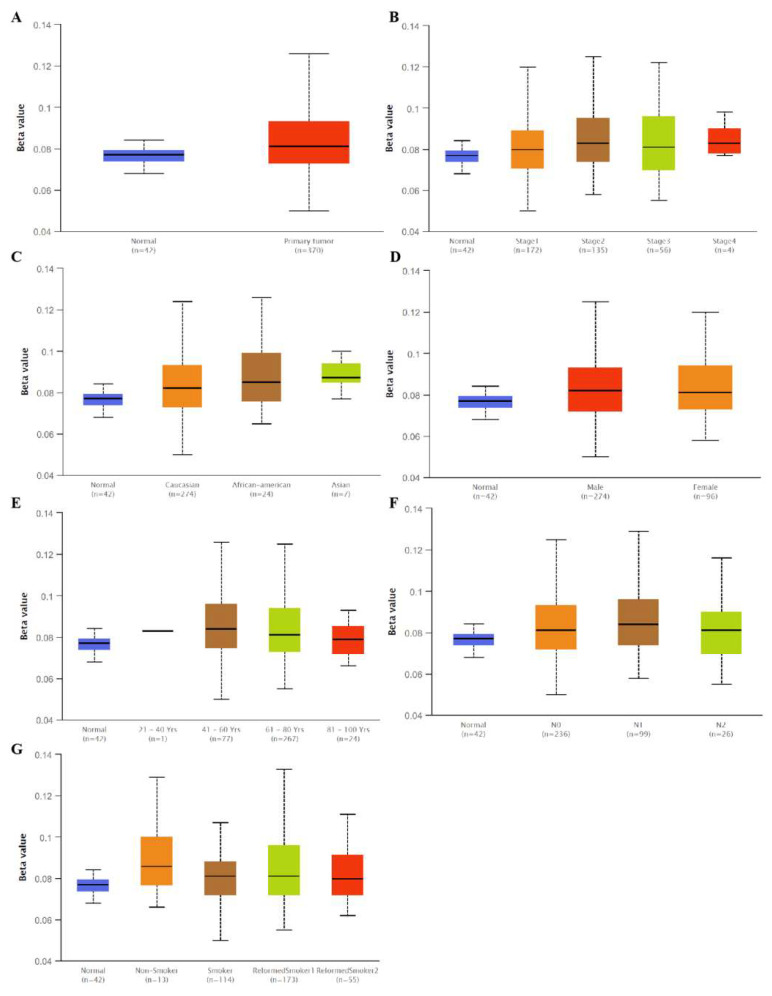
Methylation status of SLC3A2 in tumor tissues and in matched normal tissues from different subgroups of LUSC patients. (**A**) Sample type; (**B**) Stages; (**C**) Race; (**D**) Gender; (**E**) Age; (**F**) Nodel metastasis; (**G**) Smoking habits.

**Figure 5 cancers-14-05191-f005:**
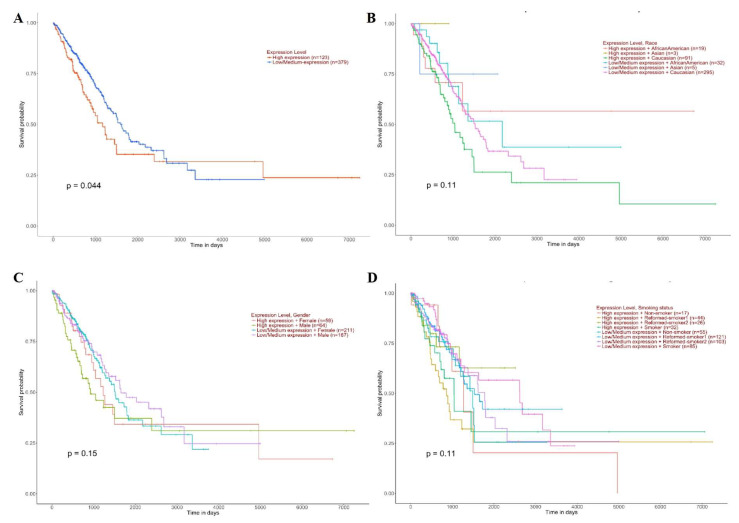
Prognostic roles of SLC3A2 in different subgroups of LUAD patients. (**A**) Sample type; (**B**) Race; (**C**) Gender; (**D**) Smoking habits.

**Figure 6 cancers-14-05191-f006:**
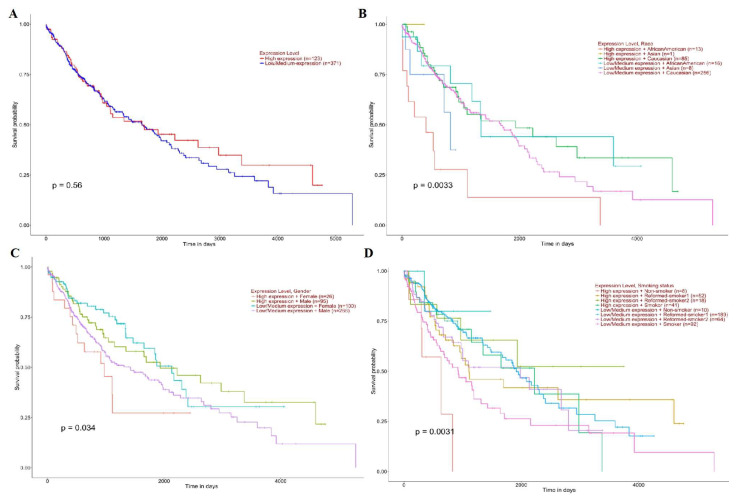
Prognostic roles of SLC3A2 in different subgroups of LUSC patients. (**A**) Sample type; (**B**) Race; (**C**) Gender; (**D**) Smoking habits.

**Figure 7 cancers-14-05191-f007:**
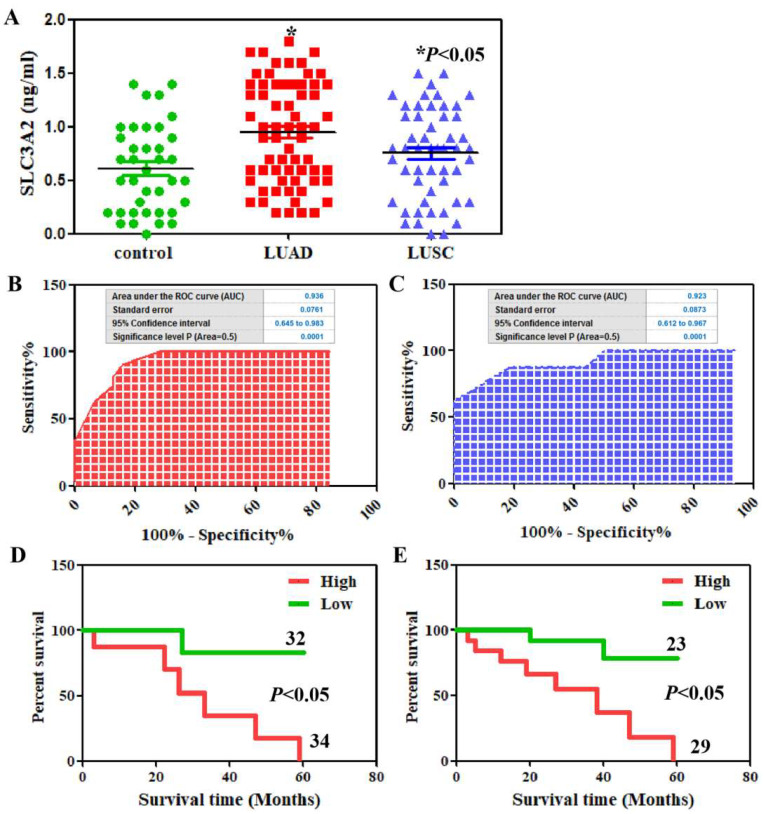
Serum SLC3A2 in LUAD and LUSC patients. (**A**) SLC3A2 serum levels from the LUAD and LUSC patients and healthy controls were detected using ELISA. ROC curve analysis for SLC3A2 diagnosis in the patients with LUAD (**B**) and LUSC (**C**). Kaplan–Meier curves of the cumulative survival rate of the patients with LUAD (**D**) and LUSC (**E**) based on their SLC3A2 serum levels.

**Figure 8 cancers-14-05191-f008:**
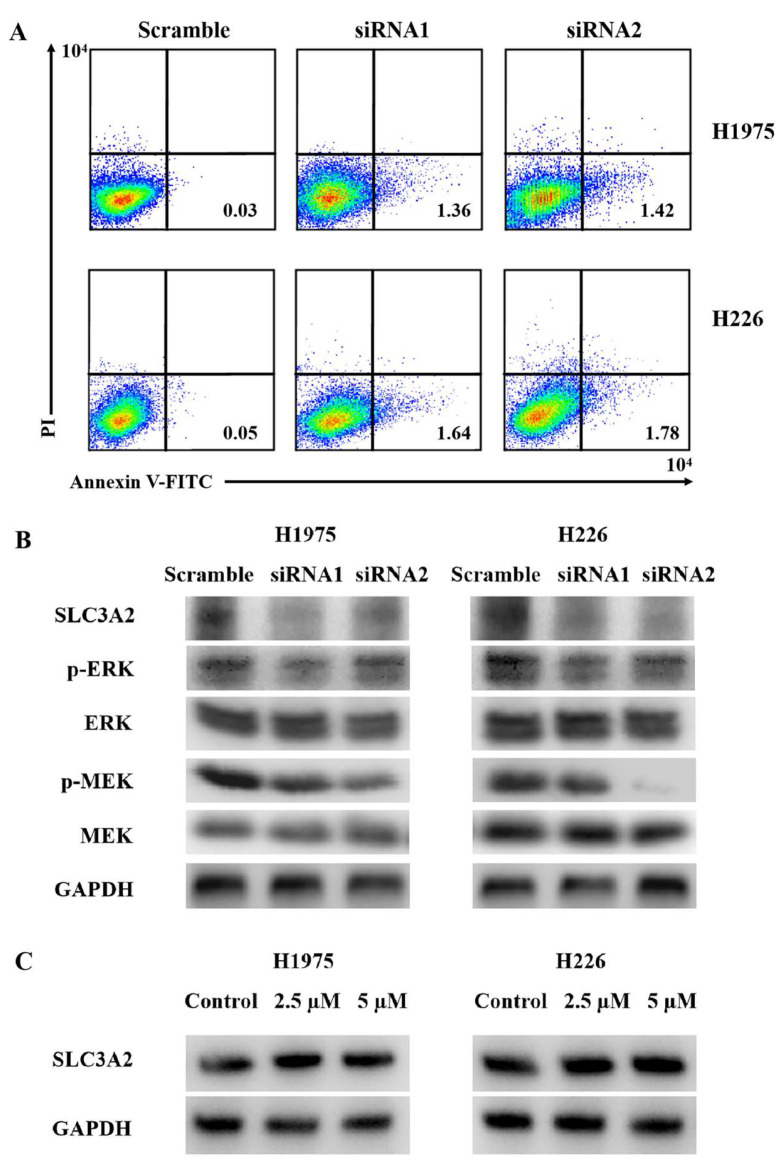
The roles and mechanisms of SLC3A2 in LUAD and LUSC cells. (**A**) The proportion of apoptotic cells (early apoptosis) was determined by double staining with Annexin-V/FITC and PI. (**B**) Western blot analysis of the ERK/MEK signaling pathway. GAPDH was used as an internal loading control. (**C**) Methylation status of SLC3A2 in LUAD and LUSC cells was checked using 5-AzaC. Uncropped WB images are shown in Appendix A.

**Figure 9 cancers-14-05191-f009:**
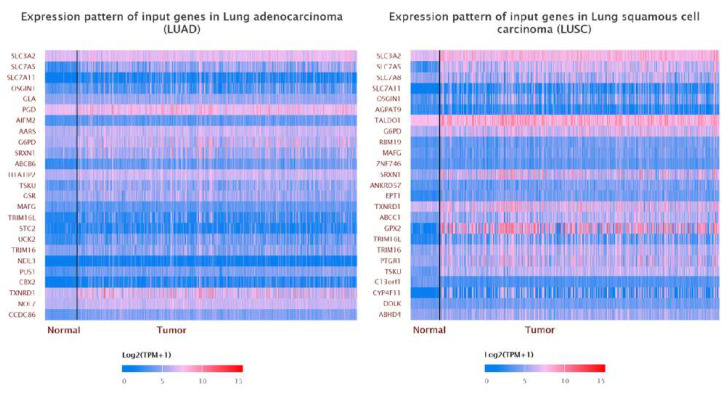
Correlation of SLC3A2 with other genes in LUAD and LUSC tissues.

**Table 1 cancers-14-05191-t001:** Relationships between serum SLC3A2 and the clinicopathological parameters of patients with LUAD or LUSC.

Clinicopathological Features	LUAD					LUSC				
	*n*	High	Low	χ^2^	*p*	*n*	High	Low	χ^2^	*p*
Sex				0.229	0.632				0.001	0.975
Male	31	15	16			25	14	11		
Female	35	19	16			27	15	12		
Age (years)				0.354	0.552				0.023	0.879
<60	23	13	10			22	12	10		
≥60	43	21	22			30	17	13		
Differentiation				**3.957**	**0.047**				**4.348**	**0.037**
Well or Moderate	23	8	15			19	7	12		
Poor	43	26	17			33	22	11		
Lymphatic invasion				**16.00**	**0.00006**				**7.364**	**0.00665**
-	25	5	20			23	8	15		
+	41	29	12			29	21	8		
Venous invasion				**14.68**	**0.00013**				**10.53**	**0.00118**
-	22	4	18			19	5	14		
+	44	30	14			33	24	9		
Tumor size				0.487	0.485				2.526	0.112
<3 cm	24	11	13			23	10	13		
≥3 cm	42	23	19			29	19	10		
pN category				0.364	0.947				0.317	0.956
pN0	11	5	6			11	6	5		
pN1	14	8	6			13	8	5		
pN2	20	10	10			18	10	8		
pN3	21	11	10			10	5	5		
Smoking status				0.121	0.727				0.001	0.974
smokers	22	12	10			27	15	12		
nonsmokers	44	22	22			25	14	11		

Abbreviations: χ^2^, Chi-square distribution.

## Data Availability

The data presented in this study are available in this article (and Appendix A).

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
