# Peer review of "Both In Situ and Circulating SLC3A2 Could Be Used as Prognostic Markers for Human Lung Squamous Cell Carcinoma and Lung Adenocarcinoma"

_cancers, 2022, doi:10.3390/cancers14215191_

Round 1

Reviewer 1 Report

Line 39: “Incidence rate and mortality of lung cancer are rising rapidly” please cite the current article which is displaying rapidly rising incidence rate of lung cancer.

Line 40-42: “The incidence rate and mortality of lung cancer are rising rapidly. The pathogenesis of lung cancer has not been fully understood, but there is evidence that the occurrence of lung cancer is related to smoking, air pollution, occupational carcinogens, diet, genetics and other factors [7-10]” you have cited articles number 7-10 in support of this statement.

However, after checking these articles, I believe these are not relevant citations. For example, authors are stating incidence rate is rising rapidly and there is evidence that the occurrence of lung cancer is related to smoking, air pollution. Reference number 10 is about segmentectomy and adherence to measures. Reference 7 is about molecular subtypes of TRP-related genes and prognosis prediction.

Please change the wording or cite different articles.

Line 43: Early detection, diagnosis and treatment of lung cancer are very important [12]. May be add for improving survival.

Line 76: “approved by the Ethics Committee of Liaoning Medical University” Is there any IRB/study number?

Line 158: “High expression of SLC3A2 indicated a poor prognosis of LUAD patients (Fig. 3A)” You may use High expression of SLC3A2 is associated with. You have used Kaplan Meier curves, therefore please also indicate medians/HR and P-value.

Line 160- 161: “The LUSC patients with high or low SLC3A2 expression have similar outcome (Fig. 6A).” I believe this is incorrect statement. The reason for saying this is because in Figure 6A the P-value is 0.56 which means the difference in curves are not statistically significant. This is different from Figure 3A where the P-value falls in statistically significant area (P-0.044). Please also document median/HR and P-value here too.

Author Response

Thanks for your insightful comments. We have revised the paper and would like to resubmit it for your consideration.

Reviewer 2 Report

thank you for your work, however there are some flaws to be addressed.

- All the figures are too small

- table 1 is not correctly displayed

- please better explain the clinical impact of your research

Author Response

(The authors gave the same response as above.)

Round 2

Reviewer 1 Report

Line 63: Materials and Methods- Please comment on TCGA database and its use in analyzing the prognostic roles in the Materials and Methods section.

Line 160: “3.3 SLC3A2 is a prognostic marker for LUAD and LUSC patients” If we are writing in line 163-164 that “There is no significant difference in survival between the LUSC patients with high and low SLC3A2 expression (p=0.56)” then what would be the basis of writing that SLC3A2 is a prognostic marker for LUSC patients (Line 160). Please try to explain the readers about the basis for writing- SLC3A2 is a prognostic marker for LUSC patients or modify the wording. If you are going to modify wording then change it in both abstract and manuscript, for e.g. Line 311, Line 160, Line 28-30.

You have also written: Line 165-167 “However, high SLC3A2 expression in female patients (p=0.0033) (Fig. 6B), African American patients (p=0.0033) (Fig. 6C), and non-smoker patients (p=0.0031) (Fig. 6D) results in a shorter survival time than in patients with low expression.” You have mentioned shorter survival time- please document median overall survival (median OS numbers) for these groups. Also, please note that are multiple co-variables in this paragraph, for eg gender, smoking status. Did we adjusted for covariables before writing that SLC3A2 is a prognostic marker for LUSC patients? In addition, please indicate whether in Line 160 (3.3), you are writing about serum or tissue SLC3A2.

Line 28, 29, 30:

 We confirmed high SLC3A2 levels in both the serum and tissue of LUAD and LUSC patients. Both serum and tissue SLC3A2 were prognostic markers for LUAD and LUSC patients.

Please read the above paragraph

Author Response

Thank you very much for your letter and advice. We have revised the paper and would like to resubmit it for your consideration.

Reviewer 2 Report

thank you.

You have addressed all my concerns.

Author Response

You have addressed all my concerns.

Thank you very much.

Round 3

Reviewer 1 Report

If you could please indicate about the need for future studies/trials to validate the findings of your study towards the end. Thanks!

Author Response

The quality of our manuscript has been greatly improved with your insightful comments. I would like to express my heartfelt thanks to you.
